# Does Exercise Training Improve Cardiac-Parasympathetic Nervous System Activity in Sedentary People? A Systematic Review with Meta-Analysis

**DOI:** 10.3390/ijerph192113899

**Published:** 2022-10-26

**Authors:** Antonio Casanova-Lizón, Agustín Manresa-Rocamora, Andrew A. Flatt, José Manuel Sarabia, Manuel Moya-Ramón

**Affiliations:** 1Department of Sport Sciences, Sports Research Centre, Miguel Hernández University of Elche, 03202 Alicante, Spain; 2Department of Sport Sciences, Alicante Institute for Health and Biomedical Research (ISABIAL), Miguel Hernandez University, 03010 Alicante, Spain; 3Department of Health Sciences and Kinesiology, Georgia Southern University—Armstrong Campus, Savannah, GA 31419, USA

**Keywords:** heart rate variability, heart rate recovery, autonomous nervous system, aerobic training

## Abstract

The aim of this study was to investigate the training-induced effect on cardiac parasympathetic nervous system (PNS) activity, assessed by resting heart rate variability (HRV) and post-exercise heart rate recovery (HRR), in sedentary healthy people. Electronic searches were carried out in PubMed, Embase, and Web of Science. Random-effects models of between-group standardised mean difference (SMD) were estimated. Heterogeneity analyses were performed by means of the chi-square test and *I^2^* index. Subgroup analyses and meta-regressions were performed to investigate the influence of potential moderator variables on the training-induced effect. The results showed a small increase in RMSSD (SMD_+_ = 0.57 [95% confidence interval (CI) = 0.23, 0.91]) and high frequency (HF) (SMD_+_ = 0.21 [95% CI = 0.01, 0.42]) in favour of the intervention group. Heterogeneity tests reached statistical significance for RMSSD and HF (*p* ≤ 0.001), and the inconsistency was moderate (*I^2^* = 68% and 60%, respectively). We found higher training-induced effects on HF in studies that performed a shorter intervention or lower number of exercise sessions (*p* ≤ 0.001). Data were insufficient to investigate the effect of exercise training on HRR. Exercise training increases cardiac PNS modulation in sedentary people, while its effect on PNS tone requires future study.

## 1. Introduction

The autonomic nervous system (ANS) is composed of the sympathetic nervous system (SNS) and parasympathetic nervous system (PNS), which operate in an inverse manner [1]. At rest, there is PNS dominance over SNS influences [2]. During exercise, cardiac PNS activity decreases and SNS activity increases [3,4,5,6]. Finally, after exercise, PNS reactivation occurs and SNS activity wanes [7].

Heart rate (HR)-based indices (e.g., HR variability [HRV] and HR recovery [HRR]), are used to estimate the ANS activity at rest and during exercise recovery, respectively [8,9]. HRV is defined as the oscillation between consecutive R-R intervals [10]. Vagal stimulation reduces HR. The vagus nerves are the main nerves of the PNS [11]. Thus, vagal activity is related to the PNS. Vagal-related HRV indices (e.g., the root mean square of the differences in successive R-R intervals [RMSSD; time-domain method], absolute values of high frequency [HF; frequency-domain method], and the standard deviation of instantaneous beat-to-beat R-R interval variability [SD_1_; Poincaré plot method]) reflect PNS modulation, while the interpretation of SNS- and PNS-mediated HRV indices (e.g., low frequency) is more controversial [12,13]. HRR is defined as the rate at which HR decreases after exercise [14]. The fast component of HRR (i.e., first minute) is mainly PNS-mediated [14]. Therefore, vagal-related HRV indices and HRR indices provide information about PNS modulation and PNS reactivation, respectively [14,15]. Nonetheless, there is evidence showing that these indices might represent independent aspects regarding cardiac PNS activity [16]. Specifically, vagal-related HRV indices reflect PNS modulation and HRR reflects PNS tone/reactivation [17].

Different methodological factors should be considered for capturing short-term (e.g., 5-min) recordings of vagal-related HRV indices for increasing the validity of the results [18,19]. First, the ANS is very sensitive to environmental influences (e.g., noise, temperature, humidity, light, and time of day) and daily physical activity and, therefore, measurements must be performed in standard conditions [11,20,21,22]. Moreover, breathing rate must be controlled to increase the validity of frequency-domain indices (i.e., HF) [23,24]. Another important consideration is the position used (e.g., supine, sitting, and standing) to increase the sensitivity of vagal-related HRV indices [25,26]. Recent studies have also suggested the use of averaged HRV values rather than single HRV values to control day-to-day HRV lability [27,28,29,30]. For HRR indices, exercise intensity, recovery mode (i.e., active or passive), and position (e.g., standing or supine) should be considered to properly interpret the results of previous studies [31,32].

Mounting evidence suggests that autonomic imbalance (e.g., decreased cardiac PNS activity) is associated with several pathological conditions [33,34] and increased mortality risk [35,36,37]. The detrimental effects related to physical inactivity on cardiovascular health have been documented over the last 70 years [38,39,40], but for the last 10 years other conditions, such as metabolic syndrome, heart failure and hypertension associated with autonomic dysfunction due to a sedentary lifestyle have emerged [41,42,43,44,45,46,47]. Autonomic dysfunction causes sympathetic hyperactivity and can have harmful effects on the cardiovascular system both directly and indirectly [48,49]. Moreover, improved cardiac PNS activity is associated with a reduction in cardiovascular risk factors over time, such as hypertension and aortic stiffness [50,51]. Thus, practical interventions that improve cardiac PNS activity are of clinical interest. Exercise training is considered a highly effective treatment for autonomic imbalance in healthy [52] and diseased people [53,54,55,56]. However, training prescription variables (e.g., exercise type [i.e., aerobic training, resistance training, and combined aerobic and resistance training], aerobic training method [i.e., high-intensity interval training (HIIT) and moderate intensity training (MIT)], and training frequency) should be taken into account to properly design exercise programmes aimed at improving cardiac PNS activity. Aerobic training is the most common exercise type used [52,53,55], while the effect of resistance training, alone or combined with aerobic training, has been less investigated. For instance, Sandercock, Bromley and Brodie [52] found that aerobic training enhances PNS modulation (i.e., HF ms^2^) in both sedentary and physically active healthy people, while the training-induced effect on PNS reactivation (i.e., HRR) was not investigated. However, the aerobic-induced effect on PNS modulation in sedentary people (subgroup analysis) was inconsistent. The controversial findings could be due to the influence of methodological factors on the sensitivity of vagal-related HRV indices and/or the influence of other potential moderator variables (e.g., training frequency, aerobic training method, and intervention length) on the improvement of PNS modulation. In this regard, there is evidence showing that HIIT enhances cardiorespiratory fitness to a greater extent than MIT in healthy [57] and diseased people [58]; whether HIIT is better than MIT for enhancing cardiac PNS activity remains unclear. Moreover, the effect of other exercise modalities (e.g., resistance training) on PNS modulation in sedentary people has not been addressed in previous systematic reviews with meta-analyses.

Therefore, the aims of the current systematic review with meta-analysis were: (a) to estimate the effect of exercise training on PNS modulation (i.e., vagal-related HRV indices) and PNS reactivation (i.e., HRR) in sedentary healthy people while accounting for methodological factors, and (b) to analyse the influence of potential moderator variables (e.g., exercise type and aerobic training method) on the improvement of cardiac PNS activity. Based on previous evidence, we hypothesise that exercise training will enhance cardiac PNS activity (i.e., PNS modulation and PNS reactivation) in sedentary people. In addition, we hypothesise that HIIT will increase cardiac PNS activity to a greater extent than MIT. Finally, due to the controversial results found, the influence of other training variables (e.g., training frequency) on the effect of exercise on cardiac PNS activity was not hypothesised.

## 2. Method

This systematic review with meta-analysis was performed following the Preferred Reporting Items for Systematic Reviews and Meta-analyses (PRISMA) guidelines [59]. The protocol of the study was prospectively registered in the PROSPERO database (CRD42021267545).

### 2.1. Data Search and Sources

Potential studies were identified via a comprehensive strategy. Electronic searches were carried out in Pubmed, Embase, and Web of Science databases from inception to February 2022 using free-text terms for the intervention (e.g., exercise) and outcomes (e.g., heart rate variability). Moreover, conference proceedings were searched in the Web of Science Core Collection. Language restrictions were not applied during this phase. The electronic search of individual databases was adapted as necessary (search terms are shown in the Appendix A). Additionally, previous systematic reviews and references of the included studies were manually searched to identify additional studies. The authors of selected studies were emailed in an attempt to identify unpublished or ongoing studies that fulfilled our selection criteria.

### 2.2. Study Selection

Eligibility criteria were established according to the PICOS (participants, intervention, comparison, outcomes, and study design) guideline as follows: (participants) sedentary healthy people regardless of sex, aged ≥ 18 years; (interventions) exercise programmes, lasting two or more weeks, based on aerobic training, resistance training, or combined aerobic and resistance training (hereafter combined training); (comparisons) non-exercise control groups; (outcomes) vagal-related HRV indices (i.e., RMSSD, HF, or SD_1_) assessed in resting conditions and/or HRR indices (e.g., HRR 1 min, HRR 2 min, and T30) and (study design) randomised and non-randomised controlled studies. Moreover, studies written in English or Spanish were included. The inclusion of studies with more than one article based on the same sample population was limited to the original publication. Two authors (A.C.-L. and A.M.-R.) independently carried out the study selection. Disagreements were settled by consensus. In cases where consensus was not reached, a third author (JM.S.) assessed the study to obtain agreement.

### 2.3. Data Extraction, Coding Study Characteristics and Potential Moderator Variables

Two authors (A.C.-L. and A.M.-R.) coded the characteristics of the included studies using a standardised data extraction form. In case of uncertainty, input from a third author (JM.S.) was obtained.

The following information was extracted from included studies: (a) study characteristics (i.e., publication year, study design, country, and sample size); (b) baseline participant characteristics (i.e., sex, male percentage, age, body weight, body mass index, oxygen peak output [VO_2_ peak], and resting heart rate [HR]); (c) intervention characteristics (i.e., exercise type, intervention length, training frequency, number of exercise sessions, aerobic training method [if applicable], intensity, setting, and other sessions details); (d) vagal-related HRV assessment characteristics (i.e., vagal-related HRV indices, power spectral density method [if applicable], outcome measure, device, assessment position, breathing control, HRV value [singe or average], and number of averaged values) and (e) HRR assessment characteristics (i.e., exercise mode, test, intensity, length, recovery type [active or passive], recovery length, position, and HRR index).

Where necessary, corresponding authors were contacted via email to request unreported information. Additionally, authors were asked about missing numerical outcome data. If no response was received, numerical values were estimated from graphically represented outcome data. We then calculated the effect size based on these estimates. Conversely, those studies where it was impossible to determine numerical data from graphical representations were excluded.

### 2.4. Methodological Quality Assessment

The tool for the assessment of study quality and reporting in exercise (TESTEX) scale was used to judge the methodological quality of the included studies [60]. This scale consists of 12 items. One point is scored if the criterion is met. Items 6 and 8 have three and two associated questions, respectively, and the maximum punctuation is 15 points. The criteria used to carry out methodological quality assessment can be found in Appendix A (see Appendix A). Based on the total scores, methodological quality was judged as excellent (12–15), good (9–11), fair (6–8), or poor (<6). Methodological quality assessment was carried out independently by two reviewers (A.C.-L. and A.M.-R.) and, if a controversial judgement was found, a third author (M.M.-R.) reviewed the specific item to reach an agreement.

### 2.5. Computation of Effect Size and Statistical Analyses

The standardised mean difference (SMD) with its 95% confidence interval (CI) was used as the ES index. The SMD was calculated by subtracting the mean change in the control group from the mean change in the intervention group divided by the pooled standard deviation at baseline and then corrected by a factor for small samples [61]. In multi-intervention studies with a shared control group, the sample size of the control groups was divided by the number of intervention groups [62]. Separate pooled analyses were carried out based on the outcome measure (i.e., RMSSD, HF, and SD_1_). The supine position was selected as a preferential among studies that assessed vagal-related HRV indices in several positions. measurement Moreover, logarithmically transformed values were selected as the preferential unit, regardless of the vagal-related HRV index. Normalised units of HF were excluded from pooled analyses. Random-effects models were used to run meta-analyses. Pooled analyses were conducted only if at least three studies were included. The SMD magnitude was classified as trivial (<0.2), small (0.2–0.6), moderate (0.61–1.2), large (1.21–2.0), or very large (>2.0) [63].

The chi-square test and the *I*^2^ statistic were used to analyse heterogeneity. A statistically significant result (*p* ≤ 0.050) for the chi-square test was indicative of substantial heterogeneity. Additionally, heterogeneity was classified as low, moderate, or high based on *I^2^* values (25, 50, and 75%, respectively). *I*^2^ index values greater than 50% were also considered of substantial heterogeneity. If substantial heterogeneity was found (*p* ≤ 0.050 and/or *I*^2^ ≥ 50%), the influence of potential moderator variables on the training-induced effect was analysed by means of subgroup analyses and simple meta-regressions for categorical (i.e., study design [randomised and non-randomised], sex [males, females, and mixed sample], assessment position [supine and seated], breathing control [yes and no], exercise type [aerobic training, resistance training, and combined training], training frequency (≤3 and >3), and aerobic training method [HIIT and MIT]) and continuous variables (i.e., intervention length and number of exercise sessions), respectively. Subgroup analyses and meta-regressions were performed if at least three or 10 analysis units had been included, respectively. The analyses of moderator variables were accomplished by assuming mixed-effects models. Egger’s test was used to carry out publication bias analysis [64]. Finally, sensitivity analyses were performed to test the robustness of our findings as follows: (a) including/excluding each individual study, and (b) excluding those studies whose methodological quality was judged as poor (TESTEX < 6). All analyses were performed using STATA software (version 16.0; Stata Corp LLC, College Station, TX, USA).

## 3. Results

The study selection process is depicted in Figure 1. Briefly, electronic database searches retrieved a total of 9201 references after deleting duplicates. Sixty-seven studies were selected after reviewing titles and abstracts, of which, 26 were included in the qualitative synthesis after reviewing the full-text [65,66,67,68,69,70,71,72,73,74,75,76,77,78,79,80,81,82,83,84,85,86,87,88,89,90] and 41 were excluded as follows: (a) participants (*n* = 12); (b) interventions (*n* = 11); (c) comparison (*n* = 7); (d) outcomes (*n* = 7); (e) data previously published (*n* = 2) and (f) language (*n* = 2). Out of all the studies included in the qualitative synthesis, seven were excluded from the quantitative synthesis due to insufficient information to calculate the ES (*n* = 5) [65,71,75,76,86] or the findings reported were not pooled (*n* = 2) [72,82]. Although efforts were made to identify unpublished studies, all included studies had been previously published in peer-reviewed journals.

The study and participant characteristics are shown in Table 1. Included studies were published from 1995 to 2022. Out of all the 26 studies included [65,66,67,68,69,70,71,72,73,74,75,76,77,78,79,80,81,82,83,84,85,86,87,88,89,90], 20 (77%) were randomised [66,68,69,70,71,72,73,74,75,76,81,82,83,84,85,86,87,88,89,90] and six (23%) were non-randomised [65,67,77,78,79,80]. Eleven studies (42%) recruited exclusively male participants [67,70,71,74,75,76,77,78,87,89,90], eight (31%) enrolled female participants [65,66,68,73,79,81,83,84], and seven (27%) recruited participants of both sexes [69,72,80,82,85,86,88]. Six studies (23%) included more than one intervention group [71,73,74,75,87,89], allowing us to define 34 analysis units. The sample size in the intervention groups varied from seven to 80 (649 participants), while the sample size in the control groups ranged from 5 to 58 (389 participants).

The intervention characteristics are summarised in Appendix A. The mean intervention length was 12.5 weeks, which ranged from 2 to 48 weeks. Sixteen studies (62%) trained thrice weekly [65,66,67,68,70,75,77,78,79,81,82,83,84,85,87,88] and only five studies (19%) carried out more than three sessions a week [69,71,76,86,89]. The total number of exercise sessions performed throughout the intervention span varied from eight to 120 sessions. Twenty-two studies (85%) performed supervised training sessions [65,66,67,68,69,70,72,73,74,75,76,77,78,80,81,83,84,85,86,87,88,90], one (4%) carried out unsupervised training sessions [71], and three (11%) did not disclose this information [79,82,89]. Out of all the 34 analysis units, 24 (70%) performed aerobic training, five (15%) carried out resistance training, and five (15%) performed combined aerobic and resistance training sessions. Out of all the 29 groups that carried out aerobic training (alone or combined with resistance training), 23 (79%) used MIT as the aerobic training method and six (21%) used HIIT. MIT session length varied between 30 and 90 min (mostly 30 min). The details of the exercise sessions can be found in Appendix A.

The details of the HRV assessment characteristics can be found in Appendix A. Twenty-five studies measured resting vagal-related HRV indices [65,66,67,68,69,70,71,72,73,74,75,76,77,78,79,80,81,83,84,85,86,87,88,89,90], of which, three (12%) used RMSSD, HF, and SD_1_ as vagal-related HRV indices [68,75,81] and nine (36%) reported RMSSD and HF [70,78,79,80,83,84,86,87,88], while one (4%), 11 (44%), and one (4%) reported exclusively RMSSD [77], HF [65,66,67,69,72,73,74,76,85,89,90], or SD_1_ [71], respectively, as the vagal-related HRV index. Out of the 22 studies that reported HF, 10 (45%) used FFT as the method to determine power spectral density [66,75,76,78,80,83,85,86,87,90], four (18%) used the auto-regressive method [69,72,74,89], two (9%) used other methods [79,84], and six (27%) did not report this information [65,67,70,73,81,88]. The outcome measures of HF can be found in Appendix A. Out of all the 25 included studies, 14 (56%) used a HR monitor to capture HRV [66,68,69,70,71,73,74,75,80,81,83,87,88,89] and 11 (44%) used an ECG [65,67,72,76,77,78,79,84,85,86,90]. Four studies (16%) measured HRV by ambulatory monitoring [69,71,85,89] and 21 (84%) by laboratory-based measures [65,66,67,68,70,72,73,74,75,76,77,78,79,80,81,83,84,86,87,88,90], of which, 11 (53%) carried out assessments in the supine position [66,68,70,73,74,77,78,81,83,87,88], three (14%) in the seated position [67,72,75], four (19%) reported HRV values measured in several positions [76,79,80,90], and three (14%) did not report this information [65,84,86]. Regarding lab-based measures, ten studies (48%) allowed the participants to breathe spontaneously during HRV assessments [66,68,70,73,74,80,81,83,87,90], five (24%) controlled the breathing rate [67,75,77,78,88], one (5%) used free and controlled conditions [76], and five (24%) did not specifically disclose this information [65,72,79,84,86].

The details of the HRR assessment characteristics and outcomes reported can be found in Appendix A.

### 3.1. Methodological Quality Assessment

The assessment of the methodological quality deemed using the TESTEX scale can be found in Appendix A. The mean ± *SD* TESTEX score was 6.7 ± 1.6 (min to max: 5 to 11). Reviewers judged five studies (19%) to have poor quality [65,67,70,71,89], 18 (69%) to have fair quality [66,68,69,72,73,74,75,76,77,78,79,80,81,82,83,84,85,90], and three (12%) to have good quality [86,87,88]. The noteworthy findings are summarised beneath. Out of all the 20 randomised studies, only three (15%) and one (5%) disclosed the method used to create the random sequence and used concealed allocation, respectively. Out of all the included studies, 19 (73%) failed to clearly disclose sample sizes at pre- and post-intervention or dropouts were higher than 15%. None of the included studies carried out intention-to-treat analysis. Only one study (4%) measured physical activity in the control group. Only four studies (17%), whose intervention length was higher than four weeks, carried out a mid-intervention assessment to maintain constant relative intensity.

### 3.2. Outcome Measures

#### 3.2.1. Parasympathetic Nervous System Modulation

Only two included studies reported enough information to calculate the ES for SD_1_ and, therefore, meta-analysis was not performed for this vagal-related HRV index. Pooled data from 12 (174 and 138 participants in the IG and CG, respectively) and 23 (470 and 231 participants in the IG and CG, respectively) analysis units showed statistical differences between the two groups in RMSSD (*p* = 0.001) and HF (*p* = 0.050) in favour of the IG. The overall SMD reached a small effect for RMSSD (SMD_+_ = 0.57 [95% CI = 0.23, 0.91]; Figure 2) and HF (SMD_+_ = 0.23 [95% CI = 0.00, 0.46]; Figure 3). Heterogeneity tests reached statistical significance for RMSSD and HF (*p* < 0.001), and moderate inconsistency was also found for both indices (*I^2^* = 68% and 63%, respectively). Therefore, heterogeneity analyses were performed for vagal-related HRV indices (i.e., RMSSD and HF).

The influence of potential moderator variables on the training-induced effect on RMSSD and HF can be found in Appendix A, respectively. The influence of position and sessions a week were not analysed for RMSSD as all pooled analysis units used the supine position and carried out three or fewer sessions a week. Subgroup analyses and simple meta-regressions showed no influence of any of the analysed variables on the training-induced effect on RMSSD (*p* > 0.050). Regarding HF, only one study measured HRV in the seated position and, therefore, the influence of the position on our findings was not analysed [67]. Subgroup analyses did not reach statistical significance (*p* > 0.050), while simple meta-regressions showed an inverse relationship between the training-induced effect on HF and the intervention length (*p* < 0.001) and the number of exercise sessions (*p* = 0.001) (see Appendix A).

The Egger’s test reached statistical significance for RMSSD (*p* = 0.004), showing a small-study effect. In contrast, no small-study effect was found for HF (*p* = 0.602). Regarding sensitivity analyses, although the conclusions were similar before and after excluding each individual study, the magnitude of the heterogeneity diminished from 68% to 0% for RMSSD, and from 63% to 15% for HF after deleting Shen and Wen [84] and Verheyden, Eijnde, Beckers, Vanhees and Aubert [90], respectively. Conclusions and heterogeneity magnitudes were the same after removing studies whose methodological quality was judged as poor. Therefore, our findings seem to be robust to the inclusion of studies with poor methodological quality.

#### 3.2.2. Parasympathetic Nervous System Reactivation

The number of included studies that reported HRR indices did not allow us to carry out pooled analyses, and their findings will be qualitatively discussed in the next section.

## 4. Discussion

The aim of the current systematic review with meta-analysis was to investigate the effect of exercise training on PNS modulation (i.e., vagal-related HRV indices) and PNS reactivation (i.e., HRR) in sedentary people while accounting for methodological factors. Moreover, we also investigated the effect of potential moderator variables (e.g., exercise type and aerobic training method) on the improvement of cardiac PNS activity. In line with our hypothesis, the findings showed that exercise training improves PNS modulation (i.e., RMSSD and HF) in sedentary people. However, it should be noted that, regardless of the vagal-related HRV index, the results of the included studies were inconsistent, which also supports our previous hypothesis. Regarding heterogeneity analyses, the results showed a greater training-induced effect on HF in studies that performed a shorter intervention or a smaller number of exercise sessions. In contrast, no influence of any of the analysed variables was found on the improvement of RMSSD. The number of included studies reporting HRR indices was inadequate for pooled analyses, which warrants future studies to increase our knowledge about the training-induced effect on PNS reactivation in sedentary people.

### 4.1. Parasympathetic Nervous System Modulation

Our findings showed an SMD increase of 0.57 (95% CI = 0.23, 0.91) in RMSSD in favour of exercise training compared to non-exercise. In agreement with our result, Pearson and Smart [91], Manresa-Rocamora, Ribeiro, Sarabia, Íbias, Oliveira, Vera-García and Moya-Ramón [56], and Picard, et al. [92] reported that exercise training enhances RMSSD in patients with chronic heart failure, coronary artery disease, and type 2 diabetes, respectively. Interestingly, Picard, Tauveron, Magdasy, Benichou, Bagheri, Ugbolue, Navel and Dutheil [92] and Manresa-Rocamora, Ribeiro, Sarabia, Íbias, Oliveira, Vera-García and Moya-Ramón [56], who also used the SMD as an ES index, found an increase of 0.62 (95% CI = 0.28, 0.95) and 0.30 (95% CI = 0.12, 0.49), respectively, in RMSSD. It should be noted that Picard, Tauveron, Magdasy, Benichou, Bagheri, Ugbolue, Navel and Dutheil [92] used within-group comparisons (i.e., pre- vs. post-intervention values) and comparisons between the changes in the intervention and control groups were not performed. The ES index used could partly explain the higher training-induced effect on RMSSD reported in patients with type 2 diabetes compared with patients with coronary artery disease. On the other hand, Pearson and Smart [91], who used the mean difference as the ES index, reported an increase of 10.44 ms (95% CI = 0.60, 20.28) in RMSSD in favour of exercise training compared to non-exercise. Regarding HF, we found an increase of 0.21 [95% CI = 0.01, 0.42]) in favour of exercise training. Similarly, Sandercock, Bromley and Brodie [52] reported an improvement of 0.43 (95% CI = 0.19, 0.67) in HF in sedentary people (subgroup analysis). It should be noted that Sandercock, Bromley and Brodie [52] included controlled and uncontrolled studies in the same pooled analysis, which could partially explain the higher training-induced effect on HF found in their meta-analysis. In line with these findings, the previous meta-analyses performed with patients also found an improvement in HF [56,91,92].

Taken together, regardless of the vagal-related HRV index used (i.e., RMSSD and HF), the results of the current and previous meta-analyses support that exercise training is a non-pharmacological treatment for improving cardiac-PNS modulation in sedentary healthy people and patients with various diseases (e.g., patients with cardiac pathologies or type 2 diabetes), which may decrease the risk of mortality [33,34,52]. Several mechanisms have been proposed to explain the effect of exercise training on PNS modulation, such as improvements in metabolic health (glucose regulation and insulin sensitivity) [93], buffering of mental stress [94], nitric oxide synthesis [95], aortic compliance and baroreflex sensitivity [87], as well as reductions in total and visceral fat mass [96], inflammation [97], oxidative stress [98], renin-angiotensin-aldosterone system activity [99], and sympathetic influence [100].

Although exercise training seems to be effective for improving cardiac-PNS modulation in sedentary people, the results of the studies included in the present review were inconsistent. Therefore, the influence of potential moderator variables on the PNS improvement was analysed. In this regard, we found no influence of exercise type (i.e., aerobic training, resistance training, and combined training) on the RMSSD and HF improvement (see Appendix A and S6). Nonetheless, most of the studies included in our review (70%), as well as in the previous meta-analyses, performed an aerobic-based exercise training programme, which limits the scope of our findings about the influence of exercise type on the training-induced effect on PNS modulation. Karavirta, Tulppo, Laaksonen, Nyman, Laukkanen, Kinnunen, Häkkinen and Häkkinen [74] and Karavirta, Costa, Goldberger, Tulppo, Laaksonen, Nyman, Keskitalo, Häkkinen and Häkkinen [73] found no differences between exercise modalities for enhancing resting HF in sedentary people. Moreover, Picard, Tauveron, Magdasy, Benichou, Bagheri, Ugbolue, Navel and Dutheil [92], who reported their findings based on the exercise type, found that aerobic training enhances vagal-related HRV indices (i.e., RMSSD and HF) in type 2 diabetic patients, while no effect was found after resistance training or combined training. Nonetheless, the number of studies that carried out resistance training was low, which limits the scope of their conclusions on the resistance training-induced effect on vagal-related HRV indices. Similarly, Bhati, et al. [101] reported in their meta-analysis that resistance training does not enhance vagal-related HRV indices in healthy people. Moreover, resistance training may decrease baroreflex sensitivity in individuals with pre-hypertension [102]. Therefore, our conclusions should be restricted to the effect of aerobic training on PNS modulation, and future studies should be performed to deeply investigate whether resistance training or combined training enhances PNS modulation to a greater extent than aerobic training in sedentary people.

There is evidence showing that HIIT is better than MIT for enhancing cardiorespiratory fitness in healthy people and patients with a wide range of pathologies [103,104]. Higher intensity is also associated with a greater improvement in HRV and aortic stiffness in sedentary adult males [87]. Consistent with this notion, Besnier, et al. [105] found that HIIT is superior to MIT for enhancing cardiorespiratory fitness and PNS modulation (i.e., HF) in patients with chronic heart failure. However, our subgroup analyses showed no differences between HIIT and MIT for enhancing cardiac-PNS modulation (i.e., RMSSD and HF) in sedentary people (see Appendix A). Moreover, none of the included studies compared the effect of HIIT and MIT on cardiac-PNS modulation, which warrants future studies to determine whether HIIT enhances vagal-related HRV indices to a greater extent than MIT. Moreover, although only two studies that performed HIIT were included, Manresa-Rocamora, Ribeiro, Sarabia, Íbias, Oliveira, Vera-García and Moya-Ramón [56] reported in their meta-analysis no differences between the two aerobic training methods for enhancing HF in patients with coronary artery disease. Therefore, it seems that MIT is a sufficient stimulus for enhancing cardiac-PNS modulation in sedentary people, and HIIT should be used to eventually increase the training stimulus and promote long-term adaptations.

Interestingly, against training principles, we found that studies that performed a shorter intervention or carried out a lower number of exercise sessions found greater HF improvement. Previous studies have highlighted the critical role of manipulating training variables (e.g., training frequency and intensity) to facilitate long-term adaptation [106,107,108,109]. In addition, we also found no influence of the training frequency on the training-induced effect on PNS modulation, which is in line with the results reported in the previous meta-analyses carried out with patients [56,92]. Nonetheless, most of the included studies performed three sessions a week. Therefore, the influence of training frequency on the training-induced effect on PNS modulation requires future study. In summary, our findings show that MIT performed three days a week for 30 min enhances cardiac-PNS modulation in sedentary people. Furthermore, proper management of the training variables should be performed to achieve long-term adaptations. However, future studies should address the influence of aerobic training the aerobic training method (i.e., HIIT vs. MIT) and training frequency, as well as the effect of resistance training and combined training, on PNS modulation in sedentary people.

Other sources of heterogeneity should be considered to try to explain the inconsistency found in the results of included studies. Our systematic review showed high heterogeneity in terms of methodological considerations in the measurement of vagal-related HRV indices. First, ten studies that carried out lab-based measures allowed the participants to breathe spontaneously, and there is evidence showing that vagal-related HRV indices, mainly HF, are influenced by the breathing rate [24]. Second, all included studies used single-time point HRV values, while previous studies suggest that average HRV values from a series of consecutive days should be used to account for and to quantify day-to-day HRV lability [30,110]. Indeed, improvements in aerobic fitness are often associated with increased stability in daily RMSSD values [111,112,113], quantified by the coefficient of variation, and more stable RHR profiles are associated with a reduced risk of cardiovascular events [114]. Third, measurements were carried out in the supine position and the seated position in studies that performed lab-based measures. However, there is evidence showing that the standing position should be used to increase PNS modulation and avoid the saturation of acetylcholine receptors [115], which seems to be paramount in participants with heightened PNS tone. Nonetheless, sedentary people show lower PNS tone than athletes [116,117,118], and other positions (e.g., supine) could also be suitable for this population. We also note that most of the included studies were judged to be of poor or fair quality, which could lead to bias. This supports the need for future high-quality clinical trials. Finally, previous studies have reported the influence of participants’ characteristics (e.g., age and cardiorespiratory fitness) on the training-induced effect in PNS modulation [119,120,121]. Nonetheless, aggregated information at the trial level was used to carry out pooled analyses and misleading relationships could be found [122], preventing us from including these variables in the heterogeneity analysis.

### 4.2. Parasympathetic Nervous System Reactivation

Although vagal-related HRV indices and HRR indices have been used to evaluate cardiac PNS activity, the underlying physiological determinants of these indices are different [16]. Moreover, previous studies have shown that the mortality risk is higher in people with lower HRR [123]. Therefore, we decided to include HRR indices in the current systematic review. To the best of the authors’ knowledge, this is the first systematic review that has addressed the effect of exercise training on PNS reactivation in sedentary people. Nonetheless, only three controlled studies fulfilled our inclusion criteria [16,122,123], of which, two used HRR 1 min to investigate the training-induced effect on PNS reactivation in sedentary people [16,123]. The scarce previous literature warrants future studies to increase our knowledge about the influence of exercise training on PNS reactivation in sedentary people.

There is evidence showing that exercise training enhances PNS reactivation in healthy [124,125] and clinical populations. For instance, Manresa-Rocamora, Ribeiro, Sarabia, Íbias, Oliveira, Vera-García and Moya-Ramón [56] and Pearson and Smart [91] found that aerobic-based exercise training enhances HRR 1 min in patients with coronary artery disease and patients with chronic heart failure, respectively. Similarly, Romagnoli, Alis, Sanchis-Gomar, Lippi and Arduini [82] found that a MIT-based training programme increases HRR 1 min in sedentary people. In contrast, Hautala, Rankinen, Kiviniemi, Mäkikallio, Huikuri, Bouchard and Tulppo [69] carried out a 2-week aerobic training programme and no differences were found between the intervention and control groups for increasing HRR 1 min, which could be due to the short intervention length. However, a paramount methodological consideration to measure PNS reactivation is the exercise intensity used before capturing HRR 1 min [14,126,127]. In this regard, Hautala, Rankinen, Kiviniemi, Mäkikallio, Huikuri, Bouchard and Tulppo [69] assessed HRR 1 min after a maximal incremental test performed on a cycle ergometer. In contrast, Romagnoli, Alis, Sanchis-Gomar, Lippi and Arduini [82] measured PNS reactivation after two submaximal constant tests (65% and 80% of maximal HR). Interestingly, Romagnoli, Alis, Sanchis-Gomar, Lippi and Arduini [82] found higher improvement in HRR 1 min after a submaximal constant test performed at 65% of maximal HR (see Appendix A). In agreement with this finding, Le Meur, et al. [128] measured HRR 1 min across a large range of exercise intensities in endurance-trained athletes and found a higher increase in HRR 1 min after lower exercise intensities, which could be due to less metaboreflex stimulation [12]. Therefore, it seems that a submaximal constant test may be more suitable than maximal testing for capturing HRR 1 min in sedentary people. Nonetheless, the low number of studies warrants future studies to investigate whether the submaximal intensity is better than the maximal intensity for measuring PNS reactivation. Finally, the three studies performed a passive recovery, most of them in the seated position. Therefore, the influence of recovery type (i.e., active and recovery) and position (e.g., standing) requires future study.

### 4.3. Strengths and Limitations

This is the first systematic review with meta-analysis that has investigated the training-induced effect on PNS modulation, as well as in PNS reactivation, in sedentary people. Moreover, the influence of training variables (e.g., aerobic training method) on the effect of exercise training on cardiac PNS activity was also assessed. Nonetheless, some limitations should also be noted. First, the number of included studies in the subgroup analyses (e.g., resistance training for RMSSD) was low, which limits the scope of these findings. Second, aggregated information was used to perform pooled analyses, and the influence of the participants’ characteristics on the effect of exercise training on cardiac PNS activity was not investigated. Third, the number of included studies did not allow us to carry out meta-analyses for PNS reactivation. Finally, all included studies have been previously published in peer-review journals.

## 5. Conclusions

Exercise training enhances PNS modulation in sedentary people. Specifically, based on our systematic review, MIT-based exercise performed three times a week for 30 min seems to be a sufficient stimulus for enhancing vagal-related HRV indices in this population. Nonetheless, this finding should be interpreted with caution due to the low methodological quality of some of the primary studies. In contrast, the effect of resistance training and combined training on cardiac-PNS modulation should be addressed in future studies. Furthermore, the effect of exercise training on PNS reactivation has been poorly investigated, and whether submaximal exercise is better than maximal exercise before measuring HRR indices should be addressed in future studies.

## Figures and Tables

**Figure 1 ijerph-19-13899-f001:**
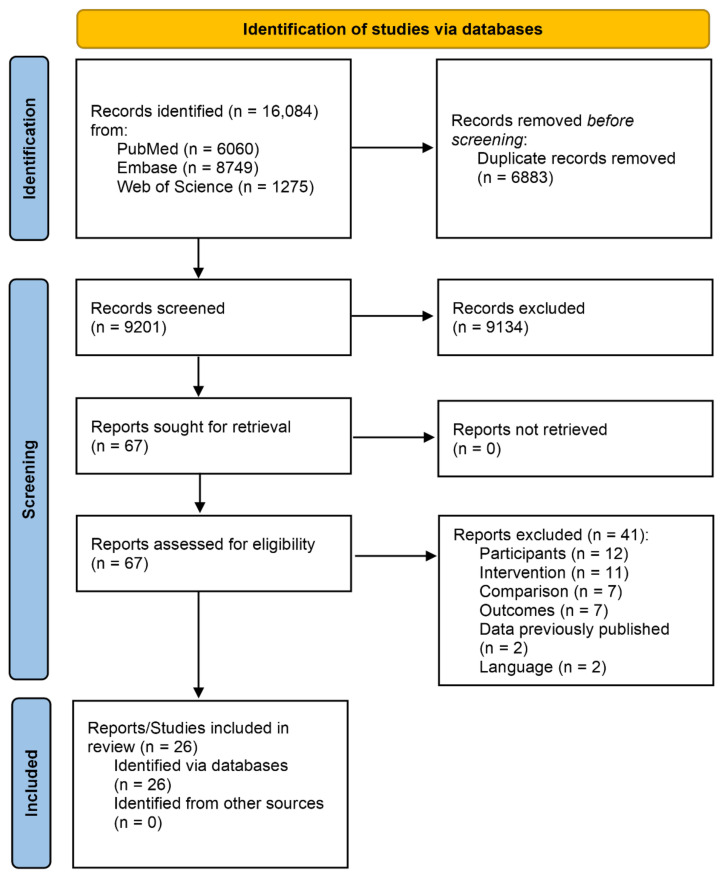
Flowchart ANS activity in sedentary people.

**Figure 2 ijerph-19-13899-f002:**
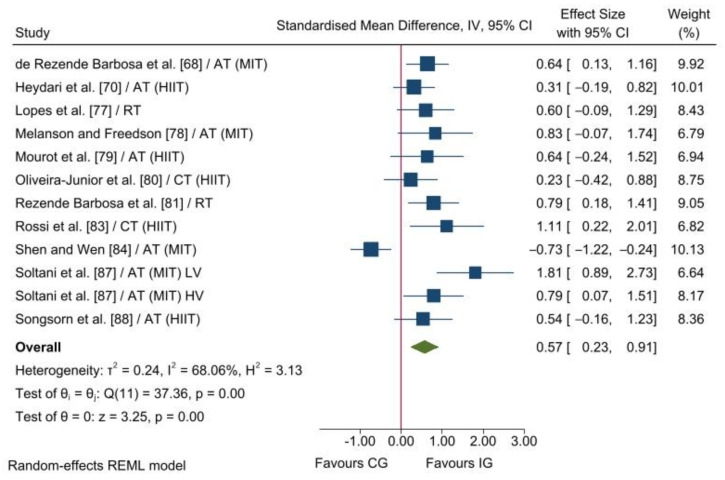
Forest plot of standardised mean difference between the intervention and control groups for RMSSD. AT, aerobic training; CG, control group; CI, confidence interval; CT, combined aerobic and training; HIIT, high-intensity interval training; HV, high volume; IG, intervention group; IV, inverse variance; LV, low volume; MIT, moderate intensity training; RT, resistance training [68,70,77,78,79,80,81,83,84,87,88].

**Figure 3 ijerph-19-13899-f003:**
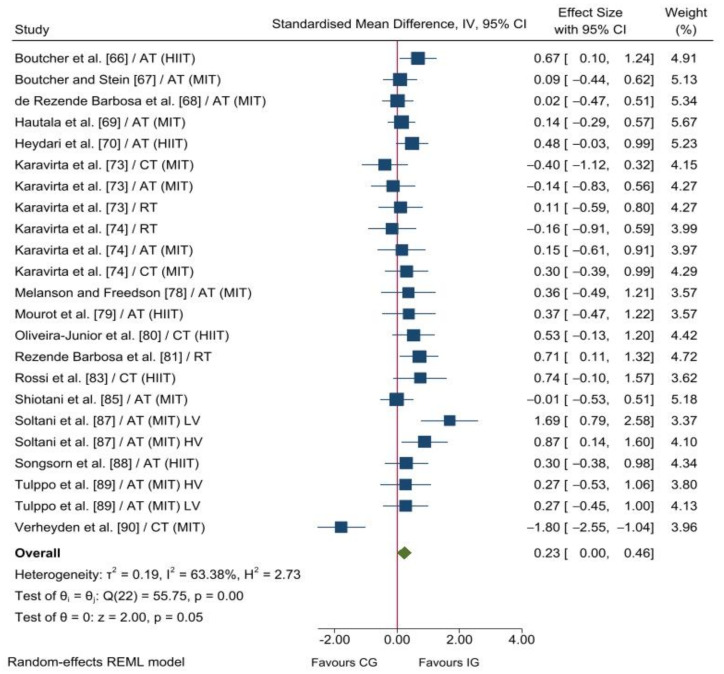
Forest plot of standardised mean difference between the intervention and control groups for HF. AT, aerobic training; CG, control group; CI, confidence interval; CT, combined aerobic and training; HIIT, high-intensity interval training; HV, high volume; IG, intervention group; IV, inverse variance; LV, low volume; MIT, moderate intensity training; RT, resistance training [66,67,68,69,70,73,74,78,79,80,81,83,85,87,88,89,90].

**Table 1 ijerph-19-13899-t001:** Study and participant characteristics.

Study (Author, Year)	Group; Exercise Type (Aerobic Method)	Study Characteristics	Patient Characteristics	
		Country; Study Design	Sex; Final Sample Size; Male Percentage; Age; Body Weight; Body Mass Index	Resting HR; VO_2_ Peak;
Audette et al. [65] 2006 *	IG; AT (MIT)	USA; non-randomised	Female; 8; 0%; 71.3 ± 4.4 years; 71.2 ± 9.5 kg; NR	NR; 23.7 ± 4.7 mL·kg^−1^·min^−1^
	CG; NA		Female; 8; 0%; 73.5 ± 5.7 years; 73.0 ± 11.3 kg; NR	NR; 26.8 ± 8.3 mL·kg^−1^·min^−1^
Boutcher et al. [66] 2013	IG; AT (HIIT)	Australia; non-randomised	Female; 16; 0%; 23.0 ± 4.0 years; 74.1 ± 12.0 kg; 27.7 ± 3.2	68.0 ± 10.4 bpm; 27.3 ± 5.2 mL·kg^−1^·min^−1^
	CG; NA		Female; 16; 0%; 21.1 ± 2.4 years; 71.3 ± 10.8 kg; 26.0 ± 2.4	70.0 ± 6.0 bpm; 29.3 ± 5.2 mL·kg^−1^·min^−1^
Lopes et al. [77] 2007	IG; RT (NA)	Brazil; non-randomised	Male; 12; 100%; 50.0 ± 4.3 years; NR; NR	68.0 ± 6.9 bpm; NR
	CG; NA		Male; 10; 100%; 48.5 ± 6.4 years; NR; NR	72.9 ± 6.2 bpm; NR
Melanson and Freedson [78] 2001	IG; AT (MIT)	USA; non-randomised	Male; 11; 100%; 36.6 ± 1.7 years; 87.2 ± 4.3 kg; 28.4 ± 1.5	64.9 ± 3.2 bpm; 38.9 ± 9.6 mL·kg^−1^·min^−1^
	CG; NA		Male; 5; 100%; 36.6 ± 4.2 years; 83.8 ± 6.6 kg; 24.9 ± 1.8	61.0 ± 4.7 bpm; 40.8 ± 7.0 mL·kg^−1^·min^−1^
Mourot et al. [79] 2005	IG; AT (HIIT)	France; non-randomised	Female; 7; 0%; 22.1 ± 3.4 years; 61.0 ± 2.9 kg; 22.2 ± NR	67.5 ± 8.7 bpm; 40.4 ± 4.8 mL·kg^−1^·min^−1^
	CG; NA		Female; 7; 0%; 22.1 ± 2.3 years; 59.6 ± 8.6 kg; 21.8 ± NR	61.5 ± 11.5 bpm; 41.5 ± 6.0 mL·kg^−1^·min^−1^
Oliveira-Junior et al. [80] 2021	IG; CT (HIIT)	Brazil; non-randomised	Both; 19; 42.1%; 36.1 ± 8.7 years; 76.5 ± 15.8 kg; 27.2 ± 4.4	NR; 34.0 ± 5.4 mL·kg^−1^·min^−1^
	CG; NA		Both; 8; 62.5%; 38.7 ± 10.6 years; 78.9 ± 14.7 kg; 27.7 ± 3.9	NR; 33.9 ± 11.5 mL·kg^−1^·min^−1^
Boutcher and Stein [67] 1995	IG; AT (MIT)	Australia; randomised	Male; 19; 100%; 46.2 ± 6.1 years; 87.1 ± 6.0 kg; NR	71.3 ± 10.0 bpm; 34.3 ± 9.8 mL·kg^−1^·min^−1^
	CG; NA		Male; 15; 100%; 45.0 ± 5.4 years; 83.9 ± 9.3 kg; NR	73.1 ± 8.1 bpm; 34.2 ± 9.4 mL·kg^−1^·min^−1^
de Rezende Barbosa et al. [68] 2018	IG; AT (MIT)	Brazil; randomised	Female; 19; 0%; 60.0 ± 4.5 years; 67.6 ± 11.6 kg; 27.3 ± 4.2	71.3 ± 8.6 bpm; NR
	CG; NA		Female; 20; 0%; 58.5 ± 4.9 years; 66.9 ± 13.2 kg; 27.6 ± 4.8	64.5 ± 8.1 bpm; NR
Hautala et al. [69] 2006	IG; AT (MIT)	Finland; randomised	Both; 80; 45%; 41.0 ± 5.0 years; NR; 25.0 ± 3.0	59.0 ± 7.0 bpm; 2.43 ± 0.7 l·min^−1^
	CG; NA		Both; 15; 60%; 41.0 ± 5.0 years; NR; 25.0 ± 2.0	57.0 ± 11.0 bpm; 2.66 ± 0.7 l·min^−1^
Heydari et al. [70] 2013	IG; AT (HIIT)	Australia; randomised	Male; 20; 100%; NR; 87.8 ± 11.7 kg; 28.4 ± 2.4	67.4 ± 9.7 bpm; 34.2 ± 4.4 mL·kg^−1^·min^−1^
	CG; NA		Male; 18; 100%; NR; 89.0 ± 12.4 kg; 29.0 ± 3.9	68.9 ± 7.7 bpm; 29.0 ± 5.5 mL·kg^−1^·min^−1^
Jelinek et al. [71] 2015 *	IG; AT (MIT) HV	Australia; randomised	Male; 15; 100%; NR; NR; NR;	NR; NR
	IG; AT (MIT) LV		Male; 19; 100%; NR; NR; NR	NR; NR
	CG; NA		Male; 16; 100%; NR; NR; NR	NR; NR
Kanegusuku et al. [72] 2015 *	IG; RT (NA)	Brazil; randomised	Both; 12; 41.7%; 64.0 ± 4.0 years; 68.3 ± 15.1 kg; 25.7 ± 4.2 kg	70.0 ± 9.0 bpm; 22.6 ± 4.9 mL·kg^−1^·min^−1^
	CG; NA		Both; 13; 15.4%; 63.0 ± 4.0 years; 68.1 ± 10.5 kg; 26.8 ± 4.7	66.0 ± 11.0 bpm; 23.3 ± 3.9 mL·kg^−1^·min^−1^
Karavirta et al. [73] 2013	IG; AT (MIT)	Finland; randomised	Female; 26; 0%; 52.0 ± 7.0 years; 66.9 ± 9.7 kg; 25.1 ± 2.7	63.0 ± 6.5 bpm; 25.3 ± 5.2 mL·kg^−1^·min^−1^
	IG; RT (NA)		Female; 26; 0%; 52.0 ± 8.0 years; 66.3 ± 9.7 kg; 24.7 ± 3.0	62.0 ± 6.5 bpm; 25.9 ± 5.4 mL·kg^−1^·min^-^
	IG; CT (MIT)		Female; 21; 0%; 49.0 ± 6.0 years; 66.2 ± 9.1 kg; 24.7 ± 3.3	62.0 ± 7.0 bpm; 27.7 ± 4.6 mL·kg^−1^·min^−1^
	CG; NA		Female; 17; 0%; 52.0 ± 8.0 years; 66.5 ± 7.5 kg; 24.1 ± 2.4	65.0 ± 5.3 bpm; 26.1 ± 5.8 mL·kg^−1^·min^−1^
Karavirta et al. [74] 2009	IG; AT (MIT)	Finland; randomised	Male; 23; 100%; 54.0 ± 8.0 years; 78.1 ± 10.0 kg; 25.1 ± 3.1 kg	61.0 ± 10.0 bpm; 32.9 ± 7.2 mL·kg^−1^·min^−1^
	IG; RT (NA)		Male; 25; 100%; 56.0 ± 6.0 years; 84.8 ± 10.0 kg; 26.8 ± 3.1	59.0 ± 9.0 bpm; 33.2 ± 6.2 mL·kg^−1^·min^−1^
	IG; CT (MIT)		Male; 29; 100%; 56.0 ± 7.0 years; 83.4 ± 11.9 kg; 26.4 ± 3.1	58.0 ± 8.0 bpm; 32.4 ± 4.2 mL·kg^−1^·min^−1^
	CG; NA		Male; 16; 100%; 54.0 ± 8.0 years; 78.0 ± 6.2 kg; 25.1 ± 1.6	54.0 ± 5.0 bpm; 35.3 ± 5.6 mL·kg^−1^·min^−1^
Kim et al. [75] 2017 *	IG; AT (MIT) 1	South Korea; randomised	Male; 12; 100%; 23.3 ± 2.2 years; NR; 24.2 ± 2.8	70.4 ± 6.6 bpm; 46.5 ± 5.0 mL·kg^−1^·min^−1^
	IG; AT (MIT) 2		Male; 12; 100%; 24.1 ± 2.2 years; NR; 23.2 ± 4.5	64.5 ± 5.7 bpm; 47.7 ± 5.4 mL·kg^−1^·min^−1^
	CG; NA		Male; 10; 100%; 23.3 ± 1.5 years; NR; 23.0 ± 3.4	70.3 ± 6.6 bpm; 46.2 ± 4.8 mL·kg^−1^·min^−1^
Lee et al. [76] 2003 *	IG; AT (MIT)	USA; randomised	Male; 12; 100%; 23.1 ± 3.0 years; 82.6 ± 16.0 kg; 26.4 ± NR	NR; 33.5 ± 3.7 mL·kg^−1^·min^−1^
	CG; NA		Male; 12; 100%; 23.1 ± 4.0 years; 79.1 ± 13.0 kg; 25.7 ± NR	NR; 33.9 ± 3.0 mL·kg^−1^·min^−1^
Rezende Barbosa et al. [81] 2016	IG; RT (NA)	Brazil; randomised	Female; 13; 0%; 23.0 ± 2.5 years; 58.3 ± 8.7 kg; 21.9 ± 2.8	NR; NR
	CG; NA		Female; 16; 0%; 20.6 ± 1.0 years; 58.4 ± 10.4 kg; 22.1 ± 3.9	NR; NR
Romagnoli et al. [82] 2018 *	IG; AT (MIT)	Spain; randomised	Both; 10; 50%; 29.7 ± 6.3 years; 67.6 ± 11.3 kg; 23.2 ± 2.7	NR; 35.4 ± 6.0 mL·kg^−1^·min^−1^
	CG; NA		Both; 10; 50%; 32.3 ± 5.7 years; 69.9 ± 13.7 kg; 23.7 ± 2.5	NR; 33.2 ± 3.5 mL·kg^−1^·min^−1^
Rossi et al. [83] 2013	IG; CT (HIIT)	Brazil; randomised	Female; 11; 0%; 62.1 ± 6.6 years; 62.4 ± 8.2 kg; NR	68.1 ± 6.4 bpm; NR
	CG; NA		Female; 6; 0%; 58.5 ± 5.1 years; 72.8 ± 17.4 kg; NR	60.7 ± 5.9 bpm; NR
Shen and Wen [84] 2013	IG; AT (MIT)	Taiwan; randomised	Female; 22; 0%; 57.9 ± 3.0 years; 55.0 ± 5.6 kg; 22.7 ± 2.2	70.9 ± 7.8 bpm; 21.5 ± 5.5 mL·kg^−1^·min^−1^
	CG; NA		Female; 22; 0%; 59.1 ± 3.9 years; 58.1± 7.7 kg; 23.7 ± 3.1	67.9 ± 8.0 bpm; 20.8 ± 6.4 mL·kg^−1^·min^−1^
Shiotani et al. [85] 2009	IG; AT (MIT)	Japan; randomised	Both; 16; 37.5%; 21.6 ± 1.7 years; NR; 20.6 ± 2.3	58.2 ± 6.5 bpm; NR
	CG; NA		Both; 19; 36.8%; 22.9 ± 1.1 years; NR; 19.7 ± 1.8	59.8 ± 7.2 bpm; NR
Sloan et al. [86] 2021 *	IG; AT (MIT)	USA; randomised	Both; 45; 46.7%; 31.2 ± 5.7 years; NR; 24.9 ± 3.8	65.3 ± 7.8 bpm; NR
	CG; NA		Both; 58; 47.5%; 31.4 ± 6.2 years; NR; 24.9 ± 3.8	67.2 ± 9.2 bpm; NR
Soltani et al. [87] 2021	IG; AT (MIT) HV	Switzerland; randomised	Male; 15; 100%; 42.5 ± 6.2 years; 73.5 ± 5.5 kg; 24.0 ± 1.3	79.2 ± 7.6 bpm; 33.3 ± 4.5 mL·kg^−1^·min^−1^
	IG; AT (MIT) LV		Male; 15; 100%; 42.2 ± 5.3 years; 73.4 ± 6.0 kg; 23.3 ± 1.5	80.1 ± 7.7 bpm; 35.2 ± 4.1 mL·kg^−1^·min^−1^
	CG; NA		Male; 15; 100%; 41.5 ± 5.6 years; 74.9 ± 6.8 kg; 24.4 ± 1.4	79.5 ± 7.7 bpm; 33.5 ± 5.0 mL·kg^−1^·min^−1^
Songsorn et al. [88] 2022	IG; AT (HIIT)	Thailand; randomised	Both; 10; 80%; 22.0 ± 0.8 years; 52.7 ± 5.9 kg; 19.5 ± 1.0	73.9 ± 13.2 bpm; NR
	CG; NA		Both; 11; 57.1%; 21.7 ± 0.8 years; 51.6 ± 5.9 kg; 19.8 ± 0.9	65.7 ± 8.9 bpm; NR
Tulppo et al. [89] 2003	IG; AT (MIT) HV	Finland; randomised	Male; 16; 100%; 35.0 ± 10.0 years; 79.0 ± 9.0 kg; 25.0 ± 2.0	54.0 ± 4.0 bpm; 42.0 ± 5.0 mL·kg^−1^·min^−1^
	IG; AT (MIT) LV		Male; 19; 100%; 35.0 ± 10.0 years; 82.0 ± 11.0 kg; 25.0 ± 3.0	55.0 ± 7.0 bpm; 41.0 ± 4.0 mL·kg^−1^·min^−1^
	CG; NA		Male; 11; 100%; 36.0 ± 11.0 years; 81.0 ± 9.0; 25.0 ± 3.0	53.0 ± 5.0 bpm; 41.0 ± 4.0 mL·kg^−1^·min^−1^
Verheyden et al. [90] 2006	IG; CT (MIT)	Belgium; randomised	Male; 14; 100%; 62.4 ± 6.1 years; 81.5 ± 3.4 kg; 26.2 ± 2.5	65.0 ± 3.0 bpm; 26.4 ± 4.5 mL·kg^−1^·min^−1^
	CG; NA		Male; 15; 100%; 64.2 ± 6.5 years; 79.8 ± 4.0 kg; 26.4 ± 3.0	66.0 ± 2.0 bpm; NR

AT, aerobic training; bpm, beats per minute; CT, combined aerobic and resistance training; CG, non-exercise control group; HIIT, high-intensity interval training; HV, high volume; HR, heart rate; IG, intervention group; LV, low volume; MIT, moderate intensity training; NA, no applicable; NR, no reported; RT, Resistance training; Sc, stretching; USA, United States of America; VO_2_ peak, peak oxygen uptake; Values are reported as mean ± standard deviation. * Excluded from the quantitative synthesis.

## Data Availability

The datasets generated from the current study are available from the corresponding author on reasonable request.

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
