# Peer review of "Does Exercise Training Improve Cardiac-Parasympathetic Nervous System Activity in Sedentary People? A Systematic Review with Meta-Analysis"

_ijerph, 2022, doi:10.3390/ijerph192113899_

Round 1

Reviewer 1 Report

This systematic review and meta-analysis study investigated whether exercise training could improve cardiac-parasympathetic nervous system activity in sedentary individuals. The topic of this study is interesting and may have some clinical significance, but the author(s) of this study explore too much information to focus the reader on the main point of the study. The paragraphs in the introduction section need to be reorganized because it is not clear why this study is so important for sedentary people.

Reviewer 2 Report

Overall, this is a well written and interesting paper that adds value to the field. I have a few minor points that will hopefully help to clarify elements of the manuscript.

Abstract

-          Please make sure the abstract meets the PRISMA abstract checklist (https://prisma-statement.org/Extensions/Abstracts)

Introduction

-          Page 1, line 39: You jump straight into using the term vagal here and do not actually explain why vagal activity is related to the parasympathetic nervous system, this would be useful for the reader. Please add that this is because of the activity of the vagus nerve innervating the heart etc (Brodal, 2016; Shaffer et al., 2014).

-          Page 1, line 41: Be careful when casually stating HF demonstrates vagally-related HRV indices as it is only HFms2 that is a measure of this - For % and normalised units, their calculation entails other parameters such a LF which means that a clear physiological origin can’t be determined for them. Having skim read the paper first this is something that is reported throughout, I would like to see the indices reported alongside any frequency variables.

-          Page 2, line 65: “improved cardiac PNS activity” – sometimes you use this phrase and other times you just use PNS activity, keep this the same throughout.

-          Page 2, line 65: “inferred from increased HRV” – this is too vague specifically state vagal activity and which HRV variables indicate this. Increased HRV is too broad and vague and we need to be specific and avoid where we can using this terminology in academic work.

-          Page 2, line 77: “(i.e. HF)” – which indices did they report?

Method

-          Page 3, line 107: The search was carried out in February… this is technically nearly 8 months out of date. I would recommend updating the search.

-          Page 3, line 107: Can you give some more detail around the PIO free-text terms? I know this is in the supplementary material – but perhaps some examples in text would be useful.

-          Pagee3, line 111: Why was forward (citation) searching not used to identify potential papers that could be included within the review?

-          Page 3, 125: What was the agreement rate between authors (inter-rater reliability)? In addition, how many disagreements where there that went to the third author and how many of those subsequently were rejected or accepted?

-          Page 4, 154: Similar question to above here (relating to the methodological assessment)

-          Page 3, 164: outcome measure of HF – were all indices of HF grouped as an outcome measure? (I have just seen that you mention this later 168 – but still what if absolute value is not presented? Did you then include another value, or was it not included). Just some more clarity needed here around HF.

Results:

Really clear and well written.  

-          Page 9, line 243-245: The 22 studies that reported HF – is this absolute power?

-          Page 9 – section 3.2.1: again just checking the HF measurement here. Please use the measure after it when reporting if it is absolute power (ms2).

Discussion:

Overarching comment about HF is the same here. Again, a well written section – I don’t have expertise in training science so I’m sorry I can’t comment more specifically on this, but the findings were discussed well.

Table S2: please report HF indices.  

References used in this review:

Brodal, P. (2016). The central nervous system – structure and function (5th ed.). Oxford University

Press.

Shaffer, F., McCraty, R., & Zerr, C. L. (2014). A healthy heart is not a metronome: An integrative review of the heart’s anatomy and heart rate  variability. Frontiers in Physiology, 5, 1040. https://doi.org/10.3389/fpsyg.2014.01040
